# Neural Ordinary Differential Equations for Modeling Epidemic Spreading

**Chrysoula Kosma**                                     *kosma@lix.polytechnique.fr*
*École Polytechnique*
*IP Paris, France*

**Giannis Nikolentzos**                                 *nikolentzos@lix.polytechnique.fr*
*École Polytechnique*
*IP Paris, France*

**George Panagopoulos**                       *george.panagopoulos@polytechnique.edu*
*École Polytechnique*
*IP Paris, France*

**Jean-Marc Steyaert**                       *jean-marc.steyaert@polytechnique.edu*
*École Polytechnique*
*IP Paris, France*

**Michalis Vazirgiannis**                               *mvazirg@lix.polytechnique.fr*
*École Polytechnique*
*IP Paris, France*

**Reviewed on OpenReview:** *https://openreview.net/forum?id=yrkJGneOuN*

## Abstract

Mathematical models of infectious diseases have long been used for studying the mechanisms by which diseases spread, for predicting the spread of epidemics, and also for controlling their outbreaks. These models are based on some assumptions and different assumptions give rise to different models. Models on social networks of individuals which capture contact patterns are usually more realistic and can more accurately model contagion dynamics. Unfortunately, computing the output of realistic models is often hard. Thus, modeling the evolution of contagion dynamics over large complex networks constitutes a challenging task. In this paper, we present a computational approach to model the contagion dynamics underlying infectious diseases. Specifically, we focus on the susceptible-infectious-recovered (SIR) epidemic model on networks. Given that this model can be expressed by an intractable system of ordinary differential equations, we devise a simpler system that approximates the output of the model. Then, we capitalize on recent advances in neural ordinary differential equations and propose a neural architecture that can effectively predict the course of an epidemic on the network. We apply the proposed architecture on several network datasets and compare it against state-of-the-art methods under different experimental settings. Our results indicate that the proposed method improves predictions in various spreading scenarios, paving the way for the extensive application of interpretable neural networks in the field of epidemic spreading. At the same time, the proposed model is highly efficient even when trained on very large networks where traditional algorithms become significantly slower.

## 1 Introduction

Spreading phenomena over complex networks are ubiquitous ranging from infectious diseases (Colizza et al., 2007) and gossip in physical networks (Chierichetti et al., 2011) to misinformation (Budak et al., 2011) and

marketing campaigns (Domingos & Richardson, 2001) on social media. Modeling such spreading phenomena over complex networks has been lying at the core of various applications over the years. Indeed, such models would allow governments and policymakers to predict and control the spread of epidemics (e.g., COVID-19) on networks of contacts (Lorch et al., 2021; Panagopoulos et al., 2021), while they would also allow social media platforms to predict and prevent the spread of rumors and misinformation (Zhao et al., 2015; Tasnim et al., 2020). Different mathematical models have been developed over the years. For instance, in epidemiology, compartmental models such as susceptible-infectious-recovered (SIR) and susceptible-infectious-susceptible (SIS), are often applied to the mathematical modeling of infectious diseases.

The outcome of a spreading process over a network is generally quantified as a node's probability of infection, for simple models like the Independent Cascade, or a quantity in compartmental models such as SIR. Several methods have been invented to derive a fast and reliable prediction of a spreading process over a given network. One can solve the system of differential equations that describes the epidemic model using computational methods (Butcher, 1996). Alternatively, the real state of the system can be approximated by simulating the spreading process multiple times in a Monte-Carlo fashion (Kempe et al., 2003; Pastor-Satorras & Vespignani, 2001). The first option is fast enough but suffers from low accuracy, while the second is more accurate but too inefficient, as it requires typically several thousands or millions of simulations for an accurate approximation. A more balanced approach, dynamic message passing, approximates the solution using dynamic equations between nodes (Karrer & Newman, 2010).

Recently, there has been an increasing interest in applying machine learning and artificial intelligence approaches to combinatorial optimization problems on networks (Dai et al., 2017; Karalias & Loukas, 2020). This approach usually involves training predictive models on instances of those problems. Once these models are trained, they can then be used for making predictions, but also for gaining insights into complex phenomena. These approaches usually rely on graph neural networks (Wu et al., 2020), a family of deep learning models that has attracted a lot of attention recently and which is particularly suited to problems that involve some kind of network structure. These models have been applied with great success to different problems such as predicting the quantum mechanical properties of molecules (Gilmer et al., 2017) and traffic prediction (Derrow-Pinion et al., 2021). Thus, graph neural networks could offer great potential to build effective data-driven dynamical models on networks. However, graph neural networks, on their own, might fail to fully capture the underlying dynamics of complex processes. For instance, in the case of mathematical models of infectious diseases such as the well-known SIR model, it might be challenging for the model to learn to predict the state of a node in a given time step. Fortunately, several of those compartmental models can be described mathematically by a set of differential equations. One can capitalize on such kind of information and incorporate structure into the learning process. This approach has already been applied to some problems (e.g., in physics) and the results indicate that it makes it easier for the model to encode the underlying dynamics (Karniadakis et al., 2021).

In this paper, we propose a novel deep neural network architecture for modeling and predicting spreading processes. We focus on the well-established susceptible-infectious-recovered (SIR) epidemiological model on arbitrary networks. In each time step, the network can be in one of $3^n$ states, where $n$ is the number of nodes of the network. The dynamics of the SIR model is described by a Markov chain on a state space of dimension $3^n$, while the time dependence of the probabilities of the states is governed by a system of $3^n$ linear ordinary differential equations. The exponential size of the system makes the analysis hard and thus, previous studies have resorted to large-scale simulations. However, for large networks, it is computationally challenging to simulate the network SIR model, and hence, for such kind of networks, little is known about the long- but also short-term evolution of the model. Instead, in this paper, we capitalize on recent advancements in the field of neural ordinary differential equations (Chen et al., 2018) and we propose a new architecture, so-called GN-ODE, to predict the probability that a node is in each one of the three states in a given time step. More specifically, we study each node individually and we employ a simpler system of differential equations which consists of $3n$ equations instead of $3^n$. Not surprisingly, by decreasing the complexity, we obtain an approximation of the exact solution. This simpler system of differential equations is integrated into a neural network model which is responsible for fine-tuning the approximate system, thus leading to more accurate predictions. The output of the neural network is computed using a black-box differential equation solver with constant memory cost. It turns out that the proposed architecture employs a message passing

mechanism similar to those of graph neural networks (GNNs) (Gilmer et al., 2017), that forms the temporal discretized approximation equations of the ODE solver, aiming to enhance their representational power in predicting epidemics spreading. To evaluate the proposed architecture, we conduct experiments on several networks of different sizes, including out of distribution testing of their generalization ability. We further investigate whether the proposed model can generalize to unseen networks by training on small networks and then evaluating the predictive performance on larger networks. Our results indicate that the proposed neural differential equation architecture outperforms vanilla GNNs in forecasting complex contagion phenomena, and thus can replace time-consuming simulations in several scenarios.

## 2 Neural ODEs for Modeling Epidemic Spreading

### 2.1 Background

**Problem definition.** Epidemics are usually studied via compartmental models, where individuals can be in different states, such as susceptible, infected, or recovered. Contact networks have been considered in modeling epidemics, as a realistic simulation of the contact process in a social context. A contact network is composed of nodes representing individuals and links representing the contact between any pair of individuals. Following previous studies, we use an individual-based SIR approach to model the spread of epidemics in networks (Youssef & Scoglio, 2011).

Let $G = (V, E)$ denote a graph (a.k.a., network) where $V$ is the set of nodes and $E$ is the set of edges. We will denote by $n$ the number of vertices and by $m$ the number of edges. The adjacency matrix $\mathbf{A} \in \mathbb{R}^{n \times n}$ of a network $G$ is a matrix that encodes edge information in the network. The element of the $i$-th row and $j$-th column is equal to the weight of the edge between vertices $v_i, v_j \in V$ if such an edge exists, and 0 otherwise. There are three states each node can belong to: (1) susceptible $S$; (2) infected $I$; or (3) recovered $R$.

The transmission of the disease is probabilistic. Thus, each edge $(v_i, v_j) \in E$ is associated with a probability that $v_i$ transmits the disease to $v_j$ in case $v_i$ becomes infected while $v_j$ is susceptible. Also, let $\beta_{ij} \in [0, 1]$ be the infection rate of edge $(i \to j)$ and $\gamma_i \in [0, 1]$ be the recovery rate of node $i$. In this work, we assume uniform infection and recovery rates, i.e., $\beta_{ij} = \beta$ for all pairs of nodes $(v_i, v_j)$ connected by an edge and $\gamma_i = \gamma$ for every node $v_i$ of the network.

In the considered model, disease spread takes place at discrete time steps $t = 1, 2, \ldots, T$. For a given network $G$ and some time step $t$, three different probabilities are associated with each node representing the probability that the node belongs to each one of the above three states. These probabilities are stored in vectors $\mathbf{s}^{(t)}, \mathbf{i}^{(t)}, \mathbf{r}^{(t)} \in \mathbb{R}^n$ for $t \in \{1, \ldots, T\}$. Given the structure of the network and some initial conditions, exactly computing those probabilities is intractable. Indeed, it has been shown that finding the probability of infection of an SIR model on a network is an NP-hard problem and that this problem is related to long-standing problems in the field of computer networks (Shapiro & Delgado-Eckert, 2012).

**Neural ODEs.** These are deep neural network models which generalize standard layer to layer propagation to continuous depth models (Chen et al., 2018). More specifically, the continuous dynamics of hidden units are parametrized using an ODE specified by a neural network:

$$\frac{d\boldsymbol{h}(t)}{dt} = f(\boldsymbol{h}(t), t, \theta)$$

where $t \in \{0, \ldots, T\}$ and $\boldsymbol{h}(t) \in \mathbb{R}^d$. Starting from the initial layer $\boldsymbol{h}(0)$, the output layer $\boldsymbol{h}(T)$ is the solution to this ODE initial value problem at some time $T$. This value can be obtained by a black-box differential equation solver. Euler's method is the simplest method for solving ODEs, among others (e.g., Runge-Kutta). For example, using Euler's/1st-order Runge-Kutta method the solution can be approximated by:

$$\boldsymbol{h}(t + s) = \boldsymbol{h}(t) + s\, f(\boldsymbol{h}(t), t, \theta)$$

where $s$ is the step size. To compute gradients with respect to all inputs of any ODE solver (Chen et al., 2010) introduce a method that scalably backpropagates through the operations of the solver. This allows training with constant memory cost as a function of depth.

## 2.2 The Proposed GN-ODE Model

As already discussed, we use an individual-based SIR approach to model the spread of epidemics in networks. Nodes represent individuals, while edges represent the contact between pairs of individuals. Unfortunately, the exact computation of the epidemic spread in a network under this model is not feasible in practice. Therefore, approximate computation schemes have been proposed, and some of them are described by a system of ordinary differential equations (ODEs) (Youssef & Scoglio, 2011). We employ the following system of ODEs:

$$
\begin{aligned}
\frac{d\mathbf{S}}{dt} &= -\beta(\mathbf{A}\,\mathbf{I}_h) \odot \mathbf{S}_h \\
\frac{d\mathbf{I}}{dt} &= \beta(\mathbf{A}\,\mathbf{I}_h) \odot \mathbf{S}_h - \gamma\mathbf{I}_h \\
\frac{d\mathbf{R}}{dt} &= \gamma\mathbf{I}_h
\end{aligned}
\tag{1}
$$

where $\mathbf{A} \in \mathbb{R}^{n \times n}$ is the adjacency matrix of the network, $\mathbf{S}, \mathbf{I}, \mathbf{R} \in \mathbb{R}^n$ are vectors that represent the three different states of the SIR model for all the nodes of the network, $\mathbf{S}_h, \mathbf{I}_h, \mathbf{R}_h \in \mathbb{R}^n$ the hidden representations of the three states and $\odot$ denotes the elementwise product. We also denote as $\beta \in [0, 1]$ the infection rate of edges and as $\gamma \in [0, 1]$ the recovery rate of nodes. By solving the above ODEs (with some initial conditions), we can approximate the spread of the epidemic in the network.

Unfortunately, the above system of ODEs might fail to capture the complex dynamics of the epidemic, thus offering solutions that are not very accurate. Thus, to overcome these limitations, we capitalize on recent advancements in neural ODEs. Specifically, we parameterize the dynamics of the individual-based SIR model using a neural network. We compute a vector for each node $v_i$ of the network, but we still require the ODEs of equation 1 to hold (this time $\mathbf{S}_h, \mathbf{I}_h, \mathbf{R}_h \in \mathbb{R}^{n \times d}$ are matrices where nodes' representations are stored in their rows). The output of the network is computed using a black-box differential equation solver.

We next give more details about the proposed model. Let $\mathbf{s}^{(0)}, \mathbf{i}^{(0)}, \mathbf{r}^{(0)} \in \mathbb{R}^n$ denote the initial conditions of the SIR instance. Hence, $\mathbf{s}^{(0)}, \mathbf{i}^{(0)}, \mathbf{r}^{(0)}$ are binary vectors and a value equal to 1 in the $i$-th component of those vectors denotes that node $v_i$ is in the corresponding state of SIR. Note that $\mathbf{s}^{(0)} + \mathbf{i}^{(0)} + \mathbf{r}^{(0)} = \mathbf{1}$ where $\mathbf{1}$ is the $n$-dimensional vector of ones. Therefore, each node initially belongs to exactly one of the three states of SIR. Those representations of the nodes are passed on to a fully-connected layer followed by the ReLU activation function, and are thus transformed into vectors of dimension $d$ (i.e., 0 and 1 integers are mapped to $d$-dimensional vectors), as follows:

$$
\begin{aligned}
\mathbf{S}^{(0)} &= \text{ReLU}\big(\mathbf{s}^{(0)}\mathbf{W}_0 + \mathbf{b}_0\big) \\
\mathbf{I}^{(0)} &= \text{ReLU}\big(\mathbf{i}^{(0)}\mathbf{W}_0 + \mathbf{b}_0\big) \\
\mathbf{R}^{(0)} &= \text{ReLU}\big(\mathbf{r}^{(0)}\mathbf{W}_0 + \mathbf{b}_0\big)
\end{aligned}
$$

where $\mathbf{W}_0 \in \mathbb{R}^{1 \times d}$ the weight matrix and $\mathbf{b}_0 \in \mathbb{R}^d$ the bias term. Thus, three vectors are associated with each node and each vector corresponds to one of the three states of the SIR model. These vectors correspond to the rows of three matrices $\mathbf{S}^{(0)}, \mathbf{I}^{(0)}, \mathbf{R}^{(0)} \in \mathbb{R}^{n \times d}$.

Then, these representations are fed to an ODE solver. The solver iteratively updates the representations of the nodes stored in matrices $\mathbf{S}^{(t)}, \mathbf{I}^{(t)}, \mathbf{R}^{(t)} \in \mathbb{R}^{n \times d}$ for $t \in \{1, \ldots, T\}$. In each iteration of the solver, first the representations of the previous iteration are further transformed using a fully-connected layer followed by the sigmoid activation function $\sigma(\cdot)$. Formally, the following updates take place:

$$
\begin{aligned}
\mathbf{S}_h^{(t)} &= \sigma\big(\mathbf{S}^{(t)}\mathbf{W}_h + \mathbf{b}_h\big) \\
\mathbf{I}_h^{(t)} &= \sigma\big(\mathbf{I}^{(t)}\mathbf{W}_h + \mathbf{b}_h\big) \\
\mathbf{R}_h^{(t)} &= \sigma\big(\mathbf{R}^{(t)}\mathbf{W}_h + \mathbf{b}_h\big)
\end{aligned}
$$

where $\mathbf{W}_h \in \mathbb{R}^{d \times d}$ the weight matrix and $\mathbf{b}_h \in \mathbb{R}^d$ the bias term. Then, the representations are re-updated based on the system of ODEs in equation 1. We need to mention that training the model requires performing

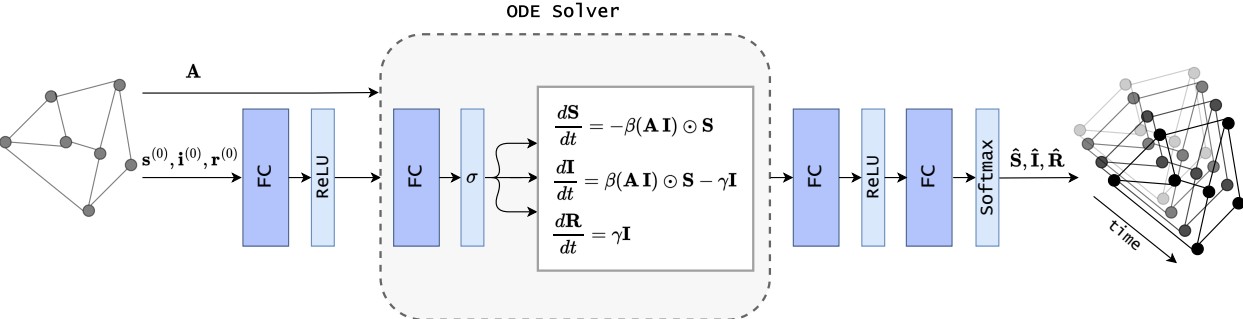

Figure 1: Overview of the proposed GN-ODE architecture.

backpropagation through the ODE solver. Even though differentiating through the operations of the forward pass is straightforward, it incurs a high memory cost and it also introduces numerical error. Following recent advancements in the field of implicit differentiation (Chen et al., 2018), we treat the ODE solver as a black box, and compute gradients using the adjoint sensitivity method (Pontryagin, 1987) which solves a second, augmented ODE backwards in time. This approach is computationally attractive since it scales linearly with problem size and has low memory requirements, while it also explicitly controls numerical error.

Once the solver has finalized its computations, the representations that correspond to the problem's discrete time steps are fed into a multi-layer perceptron (consisting of two fully connected layers) which for each node and time step outputs a 3-dimensional vector. The components of this vector correspond to the three states. Finally, the softmax function is applied to all those 3-dimensional vectors, and the emerging values can be thought of as the probabilities that a specific node belongs to state $S$, $I$ or $R$ in a given time step. These probabilities are then compared to the ground-truth probabilities that emerge from the simulations to compute the error. A high-level overview of the proposed model is given in Figure 1.

It is interesting to note that the update scheme of the ODE solver is related to a family of graph neural networks known as message passing neural networks (Gilmer et al., 2017). These models employ a message passing procedure where they update the representation of each node by aggregating information from its neighborhood. The matrix multiplication $\mathbf{A\,I}$ performed by the solver to update the states of the nodes can be seen as a form of message passing. Indeed, for each node, the output of this operation produces a vector that aggregates the representations of state $I$ of its neighbors. Then, the emerging representations are multiplied in an element-wise manner with $\mathbf{S}$. Therefore, it is evident that message passing models naturally emerge in different applications, and this perhaps justifies why these models have demonstrated great success in several problems.

## 3 Experiments

In this section, we evaluate the proposed GN-ODE model on several real-world datasets. We first present the employed datasets, the baselines and other experimental details, and then, we present and discuss the experimental results.

### 3.1 Experimental Setup

**Datasets.** We perform our experiments on real-world networks that represent social networks and are derived from online social networking and communication platforms (all datasets are publicly available). Specifically, we experiment with the following network datasets: (1) *karate* that contains social ties among the members of a University karate club; (2) *dolphins* representing a social network of bottlenose dolphins; (3) *fb-food* and (4) *fb-social* which represent the food page network of Facebook and private messages sent on a Facebook-like platform at UC-Irvine, respectively; (5) *openflights* that contains ties between two non-US-based airports and is downloaded from `Openflights.org`; (6) *Wiki-Vote*, a network created by all the voting data between administrators of Wikipedia; (7) *Enron*, an e-mail communication network; and (8) *Epinions*,

Table 1: Statistics of the 8 datasets that were employed in this study. All networks are undirected and are reduced to their largest connected component.

| Dataset | karate | dolphins | fb-food | fb-social | openflights | Wiki-Vote | Enron | Epinions |
|---|---|---|---|---|---|---|---|---|
| **#nodes** | 34 | 62 | 620 | 1,893 | 2,905 | 7,066 | 33,696 | 75,877 |
| **#edges** | 78 | 159 | 2,102 | 13,835 | 15,645 | 100,736 | 180,811 | 405,739 |
| **Transitivity** | 0.256 | 0.309 | 0.223 | 0.057 | 0.255 | 0.125 | 0.085 | 0.066 |
| **Density** | 0.1390 | 0.0841 | 0.0110 | 0.0077 | 0.0037 | 0.0040 | 0.0003 | 0.0001 |
| **Max. degree** | 17 | 12 | 134 | 255 | 242 | 1065 | 1383 | 3044 |

an online social network created from the product review website `Epinions.com`. More details about the datasets are given in Table 1. The datasets are publicly available and can be derived from the following sources: *Wiki-Vote*, *Enron*, *Epinions* are available in `https://snap.stanford.edu/data/` and the rest five datasets in `https://networkrepository.com/` (Rossi & Ahmed, 2015).

**Baseline models.** In all experiments, we compare the proposed model against three baseline methods, namely Dynamic Message Passing (DMP) (Karrer & Newman, 2010; Lokhov et al., 2015), Graph Convolution Network (GCN) (Kipf & Welling, 2017) and Graph Isomorphism Network (GIN) (Xu et al., 2019). DMP is an algorithm for inferring the marginal probabilities of stochastic spreading processes on networks. Under the individual-based SIR process, DMP is exact on trees and asymptotically exact on locally tree-like networks, while its complexity is linear in the number of edges and spreading time steps. Note that DMP is not a machine learning approach, but a purely combinatorial method. GCN and GIN are two well-established graph neural networks that have been recently applied with great success to different problems. Both models belong to the family of message passing neural networks. These architectures recursively update the representation of the nodes of a graph by aggregating information from the nodes' neighborhoods.

**Hyperparameters.** In order to select the combination of hyperparameters that leads to the best performance for each deep neural network architecture (GCN, GIN and GN-ODE), we performed grid search on a set of parameters and selected those that achieved the lowest error in the validation set. We chose learning rate from $\{0.0001, 0.001, 0.01\}$, batch size from $\{2, 4, 8, 16, 32, 64, 128\}$ and hidden dimension size for the trainable layers from $\{16, 32, 64, 128, 256, 512\}$. For larger datasets such as Wiki-Vote, Enron, Epinions we only tested the combinations of batch size and hidden dimension size that could fit into the memory of a single GPU (NVidia Quadro RTX 6000). We used the mean absolute error as our loss function and trained each architecture for 500 epochs. To make predictions, we used the model that achieved the lowest loss in the validation set. For the ODE solver in the case of GN-ODE, we used Euler's method with a step size equal to 0.5. The ground-truth values $\mathbf{s}^{(t)}, \mathbf{i}^{(t)}, \mathbf{r}^{(t)}$ were extracted after performing $10^4$ simulations for 20 time-steps.

**Evaluation metric.** We measure the mean absolute error (mae) across all nodes of all test instances, states and time steps. More specifically, the error is computed as follows:

$$mae = \frac{1}{3NnT} \sum_{i=1}^{N} \sum_{j=1}^{n} \sum_{t=1}^{T} \sum_{s \in \{S,I,R\}} |y_{i,j,t,s} - \hat{y}_{i,j,t,s}|$$

where $N$ denotes the number of test samples, $n$ the number of nodes of the graph (a.k.a., network), and $T$ the number of time steps (i. e., 20 in our setting). Furthermore, $y_{i,j,t,s}$ denotes the probability that node $j$ of the $i$-th test sample is in state $s$ in time step $t$, and $\hat{y}_{i,j,t,s}$ the corresponding predicted probability.

## 3.2 Results

We next present the experimental settings and the performance of the different models in different scenarios.

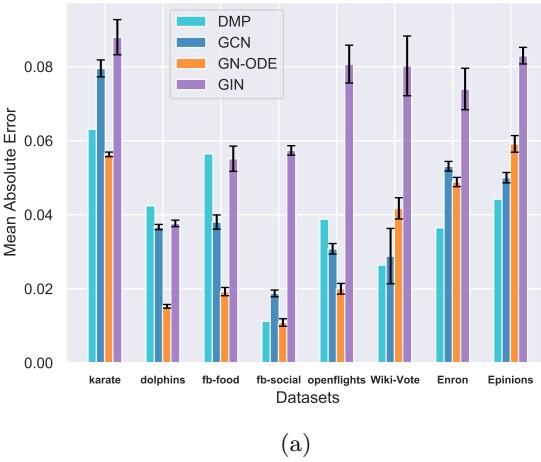 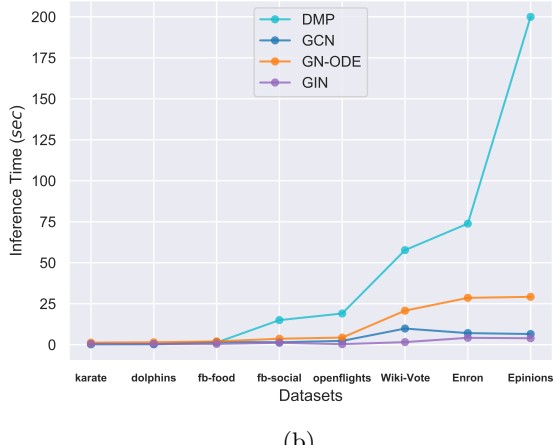

(a)                                         (b)

Figure 2: (a) Mean absolute error (lower is better) achieved by the different approaches on the test set of datasets consisting of instances of a single network structure. The values of $\beta$ and $\gamma$ for the different network instances are sampled randomly. (b) Comparison of the inference time (in sec) of the different approaches on the test set of the considered datasets.

### 3.2.1 Spreading Prediction on a Single Network

**Within distribution performance.** In the experimental results that follow, we investigate whether the different approaches can accurately estimate the spreading results of the individual-based SIR model. In these experiments, all approaches are trained and evaluated on instances of a single network.

To evaluate the performance of the different approaches, for each dataset, we created 200 samples by applying different instances of the SIR epidemic model to each network dataset. For each instance, we choose the values of hyperparameters $\beta$ and $\gamma$ of SIR randomly from $[0.1, 0.5]$ with uniform probability. This range for the hyperparameters is chosen so as to form a realistic model and to evaluate how useful each method could be in a real-world scenario (Kitsak et al., 2010). We also choose two nodes randomly with uniform probability and set them to be in the infected state $I$, while the rest of the nodes are set to be in the susceptible state $S$. To estimate the marginal probabilities, we perform $10,000$ simulations, each running for $20$ spreading time steps. The 200 samples were split into training, validation, and test sets with a $60:20:20$ split ratio, respectively.

Figure 2a illustrates the performance of the different methods. Note that each experiment is repeated 5 times, and Figure 2a illustrates the average mean absolute error along with the corresponding standard deviation. We observe that on most datasets, the proposed model outperforms the baselines. More specifically, GN-ODE is the best-performing approach on 5 out of the 8 benchmark datasets. On some datasets, the proposed model outperforms the baselines with wide margins. For instance, on the dolphins and fb-food datasets, it offers absolute improvements of $58.37\%$ and $49.41\%$ in mae, respectively, compared to the best competitor, respectively. Furthermore, on several datasets the proposed GN-ODE model achieves very low values of error, i.e., less than 0.02 which demonstrates that it can provide accurate predictions. With regards to the baselines, DMP and GCN perform comparably well in most cases, while GIN is the worst-performing method. This is an interesting result since GIN is known to be more powerful than GCN in terms of distinguishing non-isomorphic graphs (Xu et al., 2019). However, it turns out that in this task, we are more interested in estimating distributions of the states of the neighbors of a node than the exact structure of the neighborhood.

For the same set of experiments, we also demonstrate in Figure 2b the inference time of each model on the test set of the considered datasets. We can clearly observe that the inference time increases along with the size of the input networks. More specifically, the employed models show equivalent computational costs for relatively small networks such as karate, dolphins and fb-food, where the proposed GN-ODE is slightly

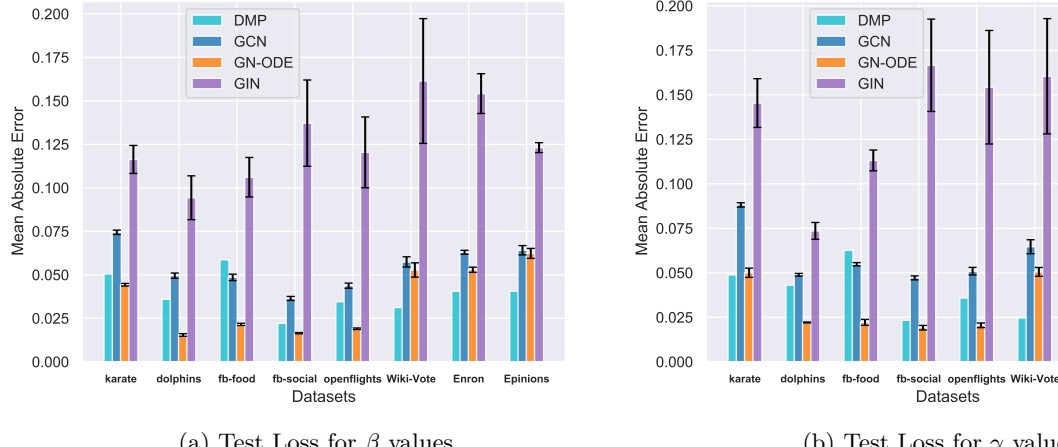

(a) Test Loss for $\beta$ values

(b) Test Loss for $\gamma$ values

Figure 3: Mean absolute error (lower is better) achieved by the different approaches on the test set of datasets consisting of instances of a single network structure. Most test instances have emerged from values of diffusion parameters $\beta$ and $\gamma$ different from those of training instances. Figures (a) and (b) illustrate the performance of the different approaches for out of distribution values of $\beta$ and $\gamma$, respectively.

slower (a few seconds) compared to the fast, in those cases, DMP. However, DMP becomes dramatically slower on larger datasets, such as Epinions, where it suffers by an approximately ten times greater inference cost compared to the proposed model. This behaviour demonstrates the necessity of the development of accurate models that remain scalable on large datasets and can be employed as alternatives to algorithms such as DMP. We also observe that the vanilla GNNs (GCN and GIN) remain quite fast even for larger networks. The inference time of the proposed GN-ODE model becomes relatively worse than that of the GNN variants, especially on the three larger networks, which can be attributed to the intermediate step used for the computations of the ODE solver, as explained in Section 2.

We also provide a visualization of the evolution of the diffusion process on the karate dataset (i.e., probabilities of infection for all the nodes of the network) in the Appendix.

**Out of distribution generalization.** Neural network models might fail to generalize to unseen data. Thus, we also perform some experiments where we study whether the different methods can accurately predict the spreading process over instances of the network that are different from the ones the methods were trained on. To achieve this, we add to the test set of a dataset, instances that emerged from values of $\beta$ and $\gamma$ that fall outside of the range of values used to train the model. In order to create the dataset, the different values of $\beta$ and $\gamma$ (from the 200 instances described above) were divided into 5 bins. Then, 80 instances sampled from bins $2, 3$ and $4$ constitute the training set. The validation and the test set both consist of some instances from bins $2, 3$ and $4$ and some instances from bins 1 and 5. Overall, the validation set contains 40 samples, while the test set contains 80 samples. Note that the training set contains instances sampled exclusively from bins $2, 3$ and $4$, while the test set mostly consists of samples from bins 1 and 5. Therefore, test instances can be considered as sampled from a different distribution compared to those of the training set.

Figure 3 illustrates the performance of the different approaches on the eight datasets. We again report the mean absolute errors across all nodes of all test instances, states and time steps. The mean absolute errors are averaged over 5 runs. We observe that for out of distribution values of both $\beta$ and $\gamma$, the GN-ODE model outperforms both GCN and GIN on all datasets. We can also see that the performance of the proposed architecture degrades on the largest networks (i.e., Wiki-Vote, Enron and Epinions) where DMP is the best-performing approach. GIN yields the worst results and achieves much higher values of mae than the rest of the methods. This might be due to the neighborhood aggregation method that this

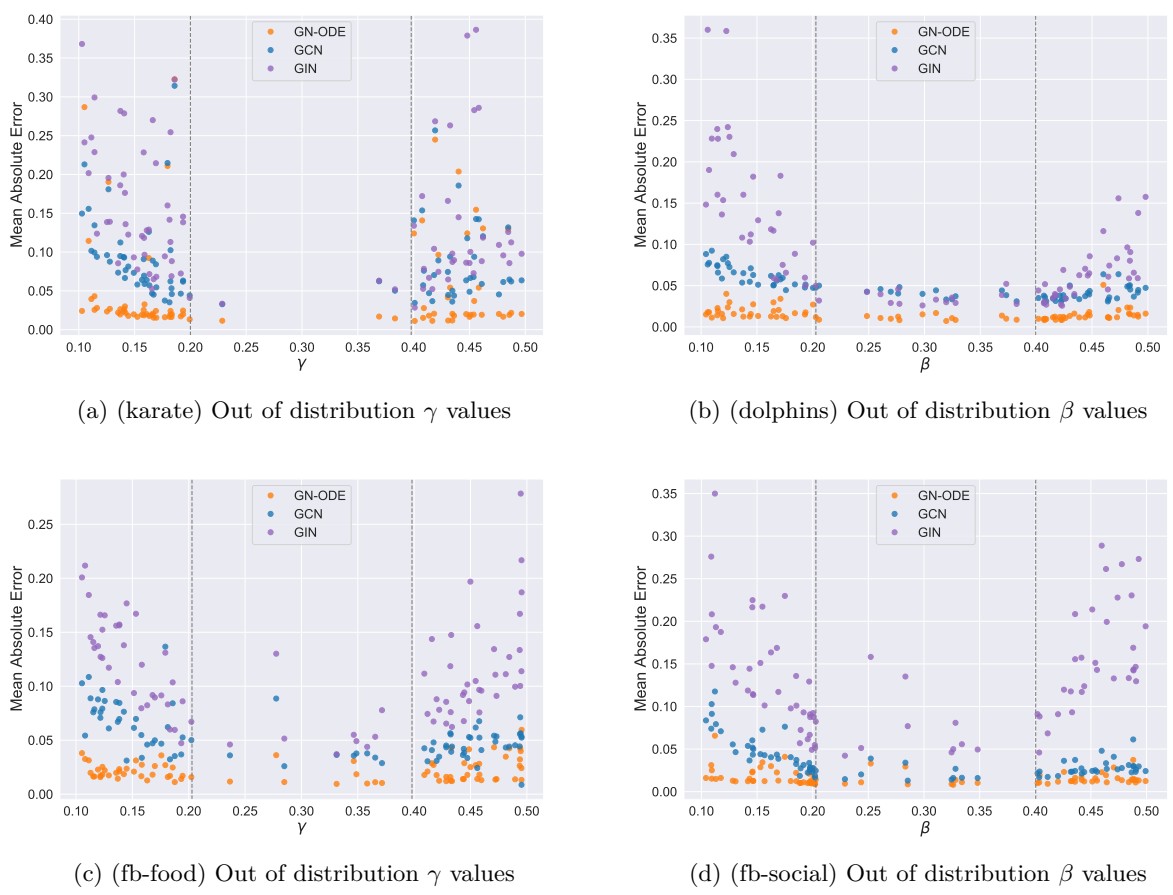

(a) (karate) Out of distribution $\gamma$ values

(b) (dolphins) Out of distribution $\beta$ values

(c) (fb-food) Out of distribution $\gamma$ values

(d) (fb-social) Out of distribution $\beta$ values

Figure 4: Mean absolute error (lower is better) achieved by the different approaches on each test sample (i. e., network) of a given dataset. Results are provided for the following four datasets: karate, dolphins, fb-food, fb-social. Each figure is associated with one dataset and one parameter ($\beta$ or $\gamma$). The out of distribution generalization performance of the different methods is evaluated. Test samples that appear in between the two dotted vertical lines correspond to test instances where values of $\beta$ and $\gamma$ were sampled from the same distribution as that of training instances. The rest of the samples correspond to instances where values of $\beta$ and $\gamma$ were sampled from different distributions than those of training instances.

neural network model utilizes (i. e., sum function). Figures 4 and 5 illustrate the error of the considered approaches for the different instances of each dataset (for clarity, we provide for each dataset a single plot illustrating the generalization performance with respect either to $\beta$ or $\gamma$). The vertical lines distinguish bins 1 and 5 (those from which test samples emerged) from bins 2, 3 and 4 (those from which training instances were sampled). The results indicate that the proposed GN-ODE model is relatively robust. In most cases, its generalization performance is similar to its within distribution performance, i. e., the obtained error for samples from bins 2, 3 and 4 is similar to the error for samples from bins 1 and 5. On the other hand, the two baseline architectures achieve lower levels of performance on instances where values of $\beta$ or $\gamma$ are different from those the models were trained on. It is interesting to note that GIN yields much higher errors for the out of distribution samples, thus the results suggest that this neural network model might not be useful in real-world scenarios. With regards to the proposed model, as already mentioned, it achieves very good levels of generalization performance on the karate, dolphins, fb-food, fb-social, and openflights datasets, while a decrease in performance occurs on the largest datasets, namely Wiki-Vote, Epinions, and Enron. Still, GN-ODE consistently outperforms the two baseline neural architectures, while the obtained errors are not prohibitive.

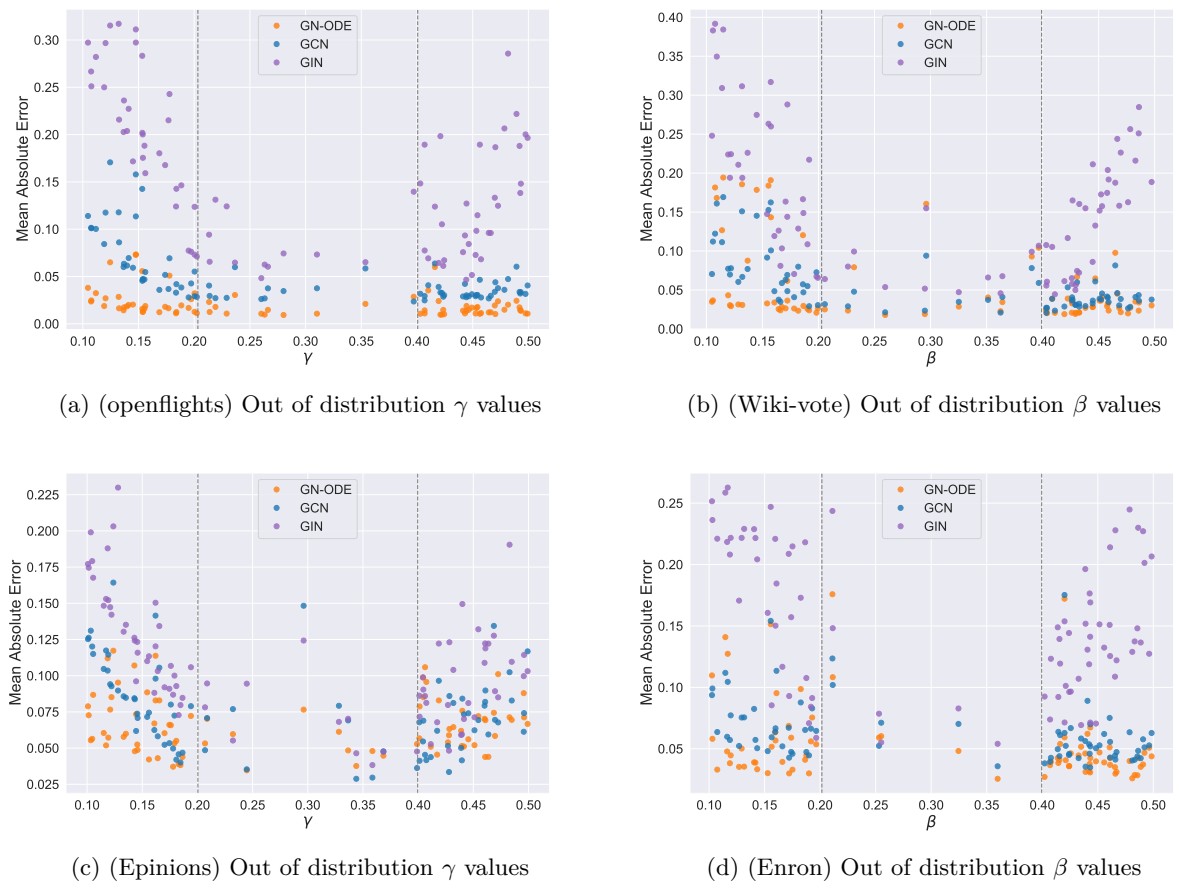

(a) (openflights) Out of distribution $\gamma$ values

(b) (Wiki-vote) Out of distribution $\beta$ values

(c) (Epinions) Out of distribution $\gamma$ values

(d) (Enron) Out of distribution $\beta$ values

Figure 5: Same as Figure 4. Results are provided for the rest of the datasets: openflights, Wiki-vote, Enron, Epinions.

### 3.2.2 Spreading Prediction on Multiple Networks

We are also interested in investigating whether a model that is trained on one or more networks can generalize to networks different from the ones it is trained on. Thus, we designed a series of experiments where the model was trained on a subset of the datasets shown in Table 1, and evaluated on some dataset that was not contained in that subset. These experiments are of very high significance since for the model to be useful in real-world problems, it is necessary that it can generalize to unseen networks. That would suggest that a model trained on some networks could be then applied to any network.

More specifically, we investigated whether models trained on karate, dolphins, fb-food fb-social, and openflights networks can accurately predict the spreading process over Wiki-Vote, Enron, and Epinions. In the case of Wiki-Vote, training was performed on instances of the dolphins, fb-food, fb-social, and openflights networks. We used 45 instances of each of those networks, i.e., 180 training samples in total. The validation and test sets contain 60 instances of the Wiki-Vote network each. In the case of the Enron and Epinions networks, training was performed on instances of the dolphins, fb-food, fb-social, openflights, and Wiki-Vote networks. Specifically, 36 instances of each of those networks were generated giving rise to 180 training samples in total. The validation and test sets both contain 60 instances of the considered network (i.e., Enron and Epinions). With regards to the rest of the hyperparameters, for each instance, $\beta$ and $\gamma$ of SIR were randomly chosen from $[0.1, 0.5]$ with uniform probability. Furthermore, for each instance, two nodes were randomly chosen with uniform probability and were set to be in the infected state $I$, while the rest

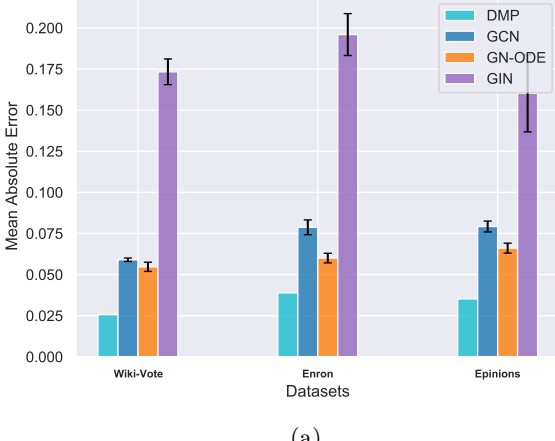
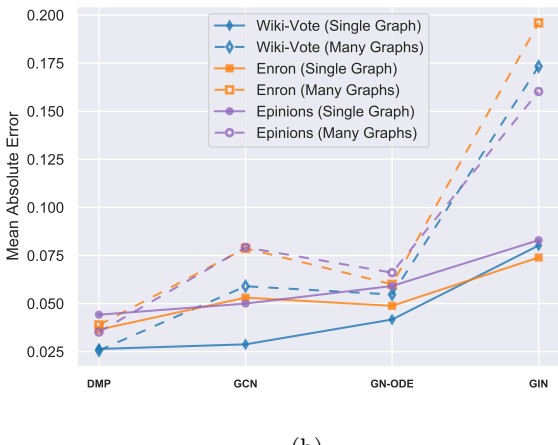

(a)        (b)

Figure 6: (a) Mean absolute error (lower is better) on the test set achieved by the different approaches when trained on instances of small networks and evaluated on instances of a large unseen network (as mentioned in the x-axis). (b) Comparison of the mean absolute error achieved by the different approaches when trained on instances of small networks and evaluated on instances of larger unseen networks (Many Graphs) vs. when both trained and evaluated on instances of the larger networks (Single Graph).

of the nodes were set to be in the susceptible state $S$. To estimate the marginal probabilities, we perform $10,000$ simulations, each running for $20$ spreading time steps.

Figure 6a illustrates the performance of the different methods on the three datasets. Once again, we report the mean absolute error achieved by each method, where we compare the predicted probabilities that nodes belong to the different states against those that emerged from the simulations. Each experiment is repeated 5 times, and besides the average mean absolute error, also standard deviations are provided. We observe that in all three experiments, the proposed GN-ODE model outperforms the baselines, i. e., the GCN and GIN models. Thus, the results suggest that the proposed model can generalize better to unseen networks than the baseline models. The GIN model fails to accurately predict the probabilities that nodes belong to the different states of SIR, thus achieving very high values of mean absolute error. On the other hand, the proposed model and GCN make more accurate predictions, and seem to be more robust since they can generalize to unseen networks.

We also investigate how the performance of the models on those three datasets compares to their performance when they are trained directly on them (i. e., results of Figures 2a and 6a combined). Figure 6b shows the results. It is clear that all models achieve better performance when instances of the network in the test set also occur in the training set. However, for the proposed model GN-ODE, we can see that the difference in performance is very small, while for the baseline models it is much higher. In the case of GIN, there is a dramatic increase in mean absolute error when the test set contains unseen networks. In the case of GCN, the increase is not that large, but still significantly greater than that of the proposed model. Overall, the results indicate that the GN-ODE model is very robust and can achieve good levels of performance even on unseen data. DMP on the other hand, can be directly applied to the test set of the unseen networks, achieving comparably better performance than the rest methods, as shown Figure 6a, with the disadvantage however of being significantly slower during inference and impractical to scale to large networks (as shown in Figure 2b).

## 4 Discussion and Conclusion

The analysis and modeling of spreading processes have been a key issue in different fields, including physics, biology and computer science, among others. For instance, predicting the course of an epidemic is of

paramount importance for governments and policymakers. Indeed, such predictions provide valuable information for adapting policies and protocols such that the spread of the disease is controlled. Mathematical models have traditionally been used to describe the underlying dynamics of different spreading phenomena. However, to capture the exact dynamics of most spreading processes, we need more realistic models which incorporate more parameters and are thus very complex. For example, most social and biological contagion processes require incorporating each individual's contact patterns in the mathematical model. Unfortunately, most real-world systems exhibit very complex connectivity patterns, thus leading to models that are hard to solve. It turns out that even the well-established SIR model on general networks is computationally intractable. Therefore, there is a need for approximation techniques which can efficiently predict the model's output.

In the past years, machine learning has emerged as a promising tool for studying physical systems and has shown great potential in providing approximate solutions to complex problems. Even though machine learning approaches can learn useful patterns directly from empirical data, recently there is a trend towards embedding the knowledge of any physical laws that govern a given dataset in the learning process. In our setting, these physical laws are described in the form of a system of ODEs. More specifically, to enhance the effectiveness of neural network models specifically for the SIR problem on networks, we incorporate knowledge on the evolution functions of the S,I,R using an approximate system of ODEs. Combining ODE solvers with neural networks has recently become an area of increasing interest for the research community. In this study, we are the first to apply task-specific neural ODEs for the SIR model, intending to advance the learning capabilities of a standard neural network model.

The obtained empirical results on a single network structure indicate that the proposed architecture outcompetes the baselines on almost all datasets and parameter settings. We need to stress that for a model to be useful, it is necessary to achieve high levels of performance even on instances of the problem that are different from the ones the model was trained on. This is, for instance, the case for most real-world datasets where training data might be available only for specific values of parameters (i.e., $\beta$ and $\gamma$) and it is thus crucial for the model to learn to generalize to previously unseen parameter values. Therefore, a great deal of emphasis was placed on testing the generalization power of the models on parameters' combinations that are not seen during training. This is because neural networks are often prone to overfitting which results in a dramatic decrease in their performance when the parameter distribution of the test set is different from that of the training set. The obtained results demonstrate that the proposed model achieves better performance than GCN and GIN for all (out of distribution) combinations of $\beta$ and $\gamma$, while it is competitive with DMP in the case of larger networks (Wiki-Vote, Enron, Epinions), where DMP achieves state-of-the-art performance at the cost of being significantly slower in terms of inference time. Besides the generalization performance with respect to the values of $\beta$ and $\gamma$, many applications require a model to be able to generalize to unseen networks. For instance, one might employ a model trained on small networks (for which ground-truth labels might be available) to make predictions for larger networks. In this setting, algorithms like Dynamic Message Passing, which are directly calculated on the test data (in our case the final large network) cannot be implemented in a way to achieve faster inference. Thus, neural network architectures that can generalize well to unseen networks can be very useful for several real-world applications. Our empirical results demonstrate that graph neural networks and especially the proposed GN-ODE model are quite successful in this task, and can thus serve as a promising approach for the modeling of spreading processes on complex networks.

Overall, even though there exist mathematical models tailored to the specificities of complex spreading phenomena, these models are usually analytically intractable. This is more evident in the case of modern large-scale applications where large networks are involved and simulation methods are inapplicable. In such settings, there is a need for approximate techniques which can accurately predict the output of the mathematical models. In this paper, we have introduced and evaluated a neural network model that is robust and scalable to large networks. The model employs prior knowledge in the form of a system of ODEs in order to increase the correctness of the function approximation. Hence, the model can make more accurate predictions and generalize well even with a small number of training examples. We believe that the proposed model can serve as a useful addition to the list of traditional approximation approaches and motivate the further development of deep learning methods for capturing the dynamics of physical systems.

## Acknowledgements

This work was supported by the French National research agency via the AML-HELAS (ANR-19-CHIA-0020) project. C.K. is supported by the IP Paris "PhD theses in Artificial Intelligence (AI)" programme by ANR.

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

## A Appendix

### A.1 Individual-based SIR Model

We next give more details about the considered individual-based SIR model. We consider a system $\Gamma$. In our setting, $\Gamma$ corresponds to a complex epidemiological system, i.e., the SIR epidemiological model. There are $3^n$ different states in total (where $n$ denotes the number of nodes of the network) and each state is denoted by $\Gamma^\alpha$ where $\alpha \in \{1, 2, \ldots, 3^n\}$. Then, the probability $P(\Gamma = \alpha)$ that the system is in state $\Gamma^\alpha$ is given by the master equations:

$$\frac{dP(\Gamma = \Gamma^\alpha)}{dt} = \sum_{\beta=1}^{3^n} \left[ R^{\beta\alpha} P(\Gamma = \Gamma^\beta) - R^{\alpha\beta} P(\Gamma = \Gamma^\alpha) \right]$$

where $R^{\beta\alpha}$ denotes the transition rate from state $\Gamma^\beta$ to state $\Gamma^\alpha$. By solving these equations, we can obtain the complete evolution of the probabilities of the states of the stochastic system $\Gamma$. However, solving these equations is only feasible for very small networks.

We suppose that within the system $\Gamma$, there exist well-defined smaller systems (i.e., subsystems). Such a set of subsystems is formed by the individuals themselves. Let $P(X_i = S)$ denote the probability that node $v_i$ is susceptible, $P(X_j = I \cap X_i = S)$ denotes the probability that node $v_j$ is infectious and node $v_i$ is susceptible. Probabilities and joint probabilities for the other states are defined in a similar manner. Then, the following system is an exact description of node probability dynamics for an SIR model on a network:

$$\frac{dP(X_i = S)}{dt} = -\beta \sum_{j=1}^{n} \mathbf{A}_{ij} P(X_j = I \cap X_i = S)$$

$$\frac{dP(X_i = I)}{dt} = \beta \sum_{j=1}^{n} \mathbf{A}_{ij} P(X_j = I \cap X_i = S) - \gamma P(X_i = I)$$

$$\frac{dP(X_i = R)}{dt} = \gamma P(X_i = I)$$

The above system is indeed exact, i.e., it gives the exact evolution of the probabilities of being susceptible, infectious or recovered during an epidemic. Unfortunately, it is not closed and thus, has no solution. We can obtain a closed system if we assume statistical independence in the states of individuals:

$$\frac{dP(X_i = S)}{dt} = -\beta \sum_{j=1}^{n} \mathbf{A}_{ij} P(X_j = I) P(X_i = S)$$

$$\frac{dP(X_i = I)}{dt} = \beta \sum_{j=1}^{n} \mathbf{A}_{ij} P(X_j = I) P(X_i = S) - \gamma P(X_i = I)$$

$$\frac{dP(X_i = R)}{dt} = \gamma P(X_i = I)$$

The above approximate set of equations (from which our model is inspired) focuses on an individual level and can be employed to evaluate the evolution of complex epidemics on networks of individuals. The accuracy of the above system depends on how much the independence assumption used to derive it holds in practice. Previous studies have shown that the above system is less accurate than more complex models (e.g., pair-based models) (Sharkey, 2008). Roughly speaking, the proposed approach uses a neural network architecture to refine the output of the above system.

### A.2 Comparison against System of ODEs

The system of ODEs of equation 1, which motivated the proposed GN-ODE model, can also be used to predict the spread of epidemics of networks as a function of time. More specifically, by solving the system for some initial values $\mathbf{s}^{(0)}, \mathbf{i}^{(0)}, \mathbf{r}^{(0)} \in \mathbb{R}^n$, we can obtain for each time step $t$ a set of vectors $\hat{\mathbf{S}}^{(t)}, \hat{\mathbf{I}}^{(t)}, \hat{\mathbf{R}}^{(t)} \in \mathbb{R}^n$ that describe the nodes' states. Note that no trainable parameters are involved in this system. We compare in

Table 2: Mean absolute error achieved by GN-ODE and simple fixed ODE system with Runge-Kutta solver, ODE-RK, on the test set of datasets consisting of instances of a single network structure. The values of $\beta$ and $\gamma$ for the different network instances are sampled randomly.

| Dataset | MODELS | |
| --- | --- | --- |
| | ODE-RK | GN-ODE |
| karate | 0.09608 | $0.05631 \pm 0.00062$ |
| dolphins | 0.10653 | $0.01527 \pm 0.00049$ |
| fb-food | 0.19109 | $0.01924 \pm 0.00111$ |
| fb-social | 0.11061 | $0.01089 \pm 0.00102$ |
| openflights | 0.16087 | $0.02000 \pm 0.00145$ |
| Wiki-Vote | 0.12287 | $0.04173 \pm 0.00287$ |
| Enron | 0.16572 | $0.04885 \pm 0.00125$ |
| Epinions | 0.15917 | $0.05915 \pm 0.00224$ |

Table 2 the proposed GN-ODE model against the solution of the system of equation 1. To solve the system, we utilized the ODE-RK method. ODE-RK follows the implementation of the SciPy package[1] and solves the fixed system of ODEs with a Runge-Kutta solver of order 5(4) (Dormand & Prince, 1980). The results reported in Table 2 highlight the poor performance of the approximate system when the representations that emerge at the different iterations of the solver are not refined by a neural network model. We can observe that for all datasets, the fixed system fails to capture the dynamics of the SIR process since it performs significantly worse compared to GN-ODE, and the rest of the methods of Figure 2.

### A.3 Visualization of the Spreading Process

We provide a visualization of the evolution of the diffusion process on the karate dataset in Figure 7. The results correspond to the experimental setting of subsection 3.2.1 and within distribution hyperparameter selection. More specifically, we illustrate the probabilities of infection (i. e., probability of a node being in state $I$) for all the nodes of the network, starting from a fixed initial set of infected nodes and fixed values of the transmission $\beta$ and recovery $\gamma$ rates, for several subsequent time steps ($t = 4$, $t = 8$ and $t = 12$). We compare the predictions obtained by applying the proposed GN-ODE model against the ground truth probabilities extracted via Monte-Carlo simulations on the test set. The color bars on the right demonstrate the ranges of the probability of infection per time step, with dark red and blue indicating the highest and lowest probabilities respectively. Not surprisingly, based on the low score in Figure 2a, it is clearly observed that the proposed GN-ODE architecture gives highly accurate predictions in comparison to the probabilities that emerge from the Monte-Carlo simulations. Due to the small size of the considered network, we notice that within a few time steps, many nodes become infected. In contrast, others obtain low probabilities of infection, probably by transitioning to the recovered set.

### A.4 Out of Distribution Generalization - Complementary Figures

We provide complementary results in Figure 8 for the out of distribution performance of the different methods for each dataset and different values of parameters $\beta$, $\gamma$. These results complement those of Figures 4 and 5. The results show the error of the considered approaches for different instances of each dataset, including samples outside of the ranges of the parameters $\beta$ or $\gamma$ in the training set. Following the observations made for Figures 4 and 5, in the plots for each dataset and the remaining out of distribution parameters $\beta$ or $\gamma$ in Figure 8, the proposed GN-ODE method seems to have comparatively more robust performance when generalizing to unseen data compared to the other GNN models.

---

[1]https://docs.scipy.org/doc/scipy/reference/integrate.html

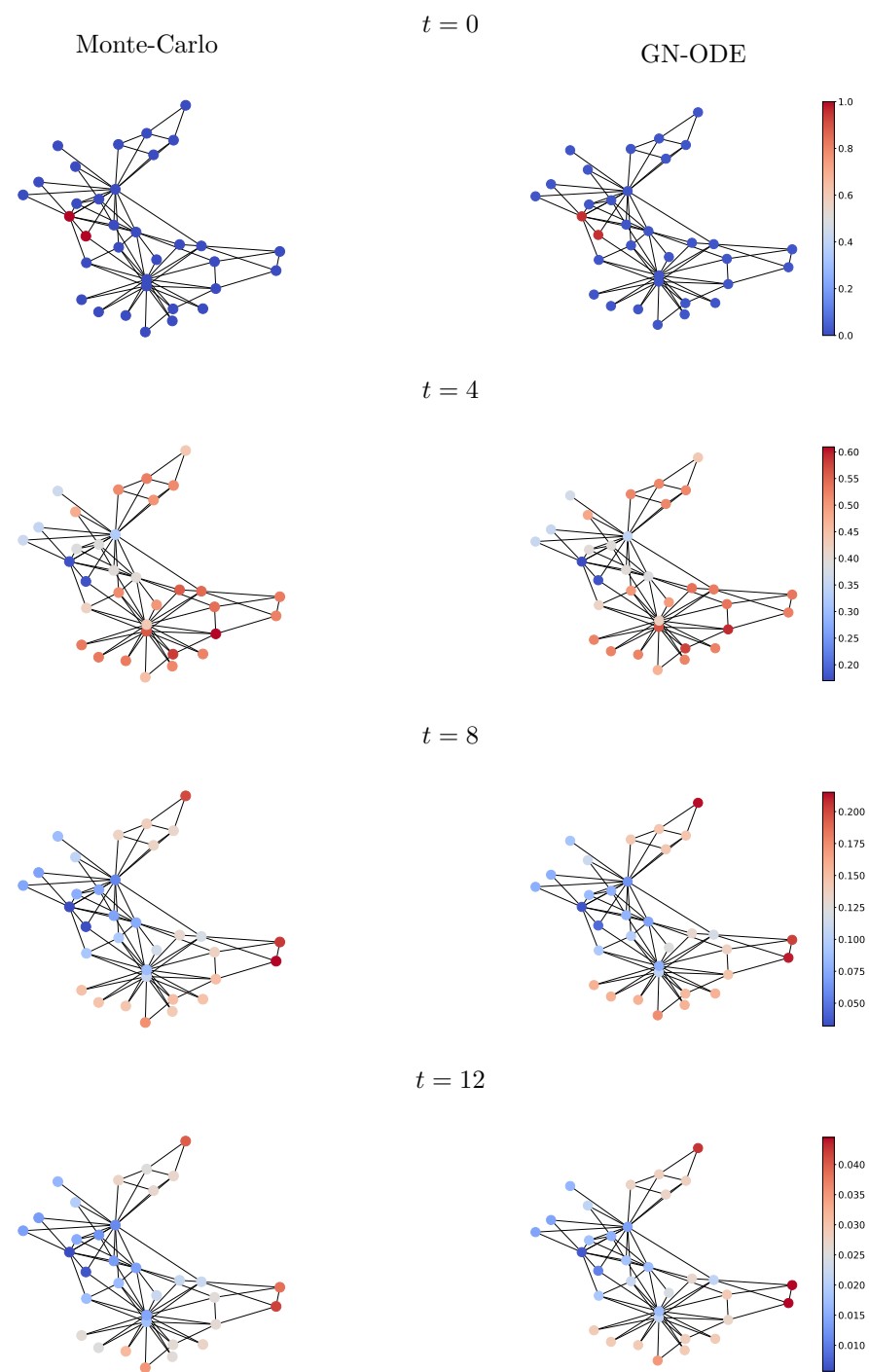

Figure 7: Visualization of the evolution of infection over time on the karate dataset. Given the initially infected nodes (with red at $t = 0$), we compare the predictions (probability that a node is in state $I$) of the proposed GN-ODE model (right) against the ground truth probabilities obtained through Monte-Carlo simulations (left).

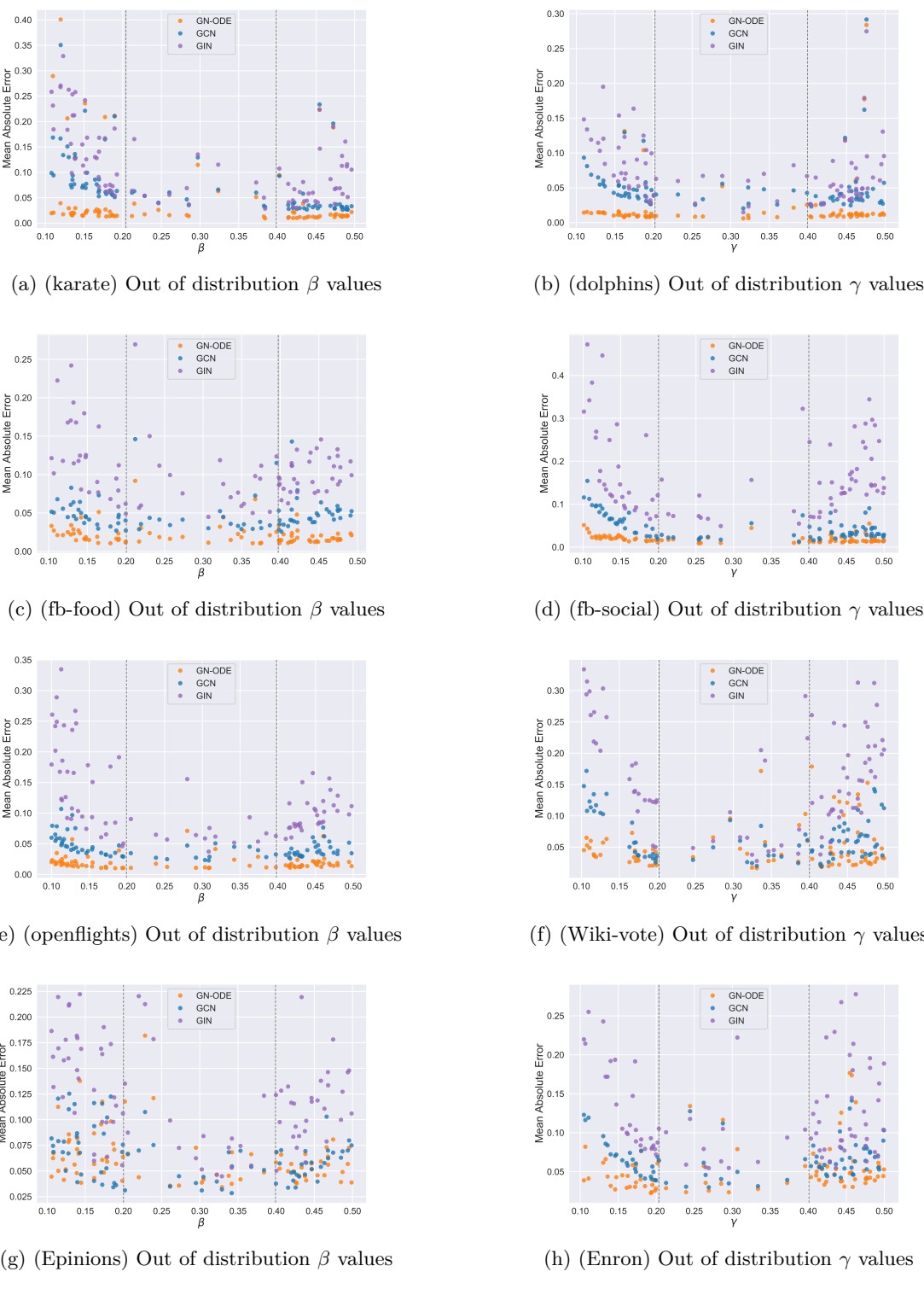

(a) (karate) Out of distribution $\beta$ values

(b) (dolphins) Out of distribution $\gamma$ values

(c) (fb-food) Out of distribution $\beta$ values

(d) (fb-social) Out of distribution $\gamma$ values

(e) (openflights) Out of distribution $\beta$ values

(f) (Wiki-vote) Out of distribution $\gamma$ values

(g) (Epinions) Out of distribution $\beta$ values

(h) (Enron) Out of distribution $\gamma$ values

Figure 8: Mean absolute error (lower is better) achieved by the different approaches on each test sample (i.e., network) of a given dataset. Each figure is associated with one dataset and one parameter ($\beta$ or $\gamma$). The out of distribution generalization performance of the different methods is evaluated. Complementary figures for each dataset and the out of distribution parameter ($\beta$ or $\gamma$) are shown in Figures 4 & 5.

