# OpenReview forum: "Neural Ordinary Differential Equations for Modeling Epidemic Spreading"
_TMLR — Accepted by TMLR_

### Review · Reviewer_eMqf · 2023-04-16

**Summary Of Contributions:**

The authors propose a neural ordinary differential equation (ODE) solver for a susceptible-infectious-recovered (SIR) model of epidemics on networks. SIR models and variants are commonly used to model the spread of disease across a network of human contacts. Such models generally cannot be solved analytically except in special cases, so researchers generally rely on Monte Carlo simulations to compute the outputs. For large networks, such simulations are quite slow, and the neural ODE-based approach the authors propose aims to predict the SIR model outputs quickly.

The main contributions I observe are as follows:
- Proposing a neural ODE-based solution (GN-ODE) for epidemic models, which is novel to the best of my knowledge.
- Extensive experimental comparison demonstrating the utility of the proposed approach compared to other approximate methods and other graph neural networks (GNNs).

**Audience:**

Yes

**Broader Impact Concerns:**

No statement is included, and I don't believe that one is necessary.


**Claims And Evidence:**

Yes

**Requested Changes:**

Major issues:
- The authors should compare directly against an ODE solver applied to the ODE-based approximation of Youssef & Scoglio (2011) in equation (1). If this is not a valid comparison, then explain why.
- The authors should include DMP also in Figure 6(a) and possibly Figure 6(b). Since DMP does not require training, I surmise that its error should be the same as in Figure 2(a). This presents a fairer comparison, showing that the accuracy of GN-ODE when generalizing to new networks is still somewhat weak, despite being better than the other GNN-based baselines. With that being said, I find that it is still useful given that its inference time is much faster than DMP.

Minor issues:
- It may be useful to the reader if the authors explained why the system of ODEs in (1) is an approximation to the actual SIR model and why they expect it to be a poor one. For example, is it due to the fixed scalar hidden representations $\mathbf{S}_h, \mathbf{I}_h, \mathbf{R}_h$?
- It looks to me like Figures 4 and 5 are showing the same type of results, just on different datasets. These should probably be merged into a single figure taking the whole page rather than two figures split across the top halves of two pages. The other cases for Figures 4 and 5 should be shown in an appendix. This would contain the other case  for each dataset not shown in Figures 4 and 5, either out of distribution $\beta$ values or out of distribution $\gamma$ values.

**Strengths And Weaknesses:**

Strengths:
- Proposed GN-ODE is useful, providing a solver that is faster than dynamic message passing (DMP) approach and more accurate than some basic GNN-based baselines.
- GN-ODE is still quite accurate when generalizing to different SIR model parameters than it was trained on.
- Paper is well written and of a good length for the scope of the contribution.

Weaknesses:
- The authors do not compare against the ODE-based approximation of Youssef & Scoglio (2011), which looks to be applicable to the same experimental setting they consider. This is odd to me, because they claim that this pure ODE-based approach cannot capture the complex dynamics of the epidemic while GN-ODE potentially could.
- Training time is not considered. One of the comparisons is against the DMP approach that does not require training and thus has much slower inference time. For a more complete comparison, the training time should be included also.
- Accuracy relative to DMP when generalizing to new networks is somewhat weak.

---

> ### Author Response · Authors · 2023-05-08
> **Reply to Reviewer eMqf**
>
> We would like to thank the Reviewer for the very concrete and helpful comments provided on our work.
> We will mainly focus on the weaknesses and requested changes for improving the manuscript.
>
> **[R]: Baseline fixed ODE system - Results:**
>
> Indeed, it is possible to directly use the system of ODEs of Youssef \& Scoglio (2011) (i.e., Equation (1)) to compute the spread of epidemics on the considered networks. Some preliminary experiments we conducted at the beginning of this study demonstrated the poor performance of this system of ODEs.  As suggested by the reviewer, we report in the revised manuscript (Table 2 in the Appendix) the performance of the system of ODEs of Equation (1) with no learnable components solved by the Runge-Kutta algorithm. The results clearly indicate the poor performance achieved by this model. Specifically, we observe that this model is even outperformed by GIN (the previously worst-performing method). Results are also mentioned in the table below.
>
>
> |  **Dataset**                                | **ODE-RK**                     |       **GN-ODE**       |
> |----------------------------------|-------------------------------------|-------------------------------------|
> | karate                           | 0.09608                             | 0.05631 $\pm$ 0.00062 |
> | dolphins                         | 0.10653                             | 0.01527 $\pm$ 0.00049 |
> | fb-food                          | 0.19109                             | 0.01924 $\pm$ 0.00111 |
> | fb-social                        | 0.11061                             | 0.01089 $\pm$ 0.00102 |
> | openflights                      | 0.16087                             | 0.02000 $\pm$ 0.00145 |
> | Wiki-Vote                        | 0.12287                             | 0.04173 $\pm$ 0.00287 |
> | Enron                            | 0.16572                             | 0.04885 $\pm$ 0.00125 |
> | Epinions                         | 0.15917                             | 0.05915 $\pm$ 0.00224 |
>
> **[R]: The fixed ODE system as an approximation:**
>
> In the considered SIR epidemiological model, there are $3^n$ different states in total (since there are $n$ nodes and three different states). The probability that the epidemiological system is in a specific state is given by a master equation. This probability depends on the contribution from all other states to the given state and thus is tractable only for very small networks (where $n$ is very small) [1]. If we assume that within the epidemiological system there exist well-defined smaller systems, we can derive subsystems such as those formed by the individuals themselves. Each subsystem can be described by three equations. Therefore, there exist $3n$ equations in total. Unfortunately, even though this set of  subsystems is exact, it is not closed and thus, has no solution [1]. The system of Youssef \& Scoglio [2] corresponds to a relaxation of the above system where statistical independence in the states of individuals is assumed. The emerging system is closed but not exact. Its accuracy depends on how much the independence assumption used to derive it holds in practice. Its accuracy has been experimentally investigated in previous studies [1,2]. More details about the different systems are given in the Appendix of the revised manuscript.
>
> **[R]: Why is GN-ODE more flexible:**
>
> As already discussed, the system of Youssef \& Scoglio [1] is not exact. It assumes statistical independence in the states of individuals and its accuracy depends on how much the independence assumption holds in practice. The main motivation behind the GN-ODE model is the use of a neural network to produce more accurate predictions. The proposed model refines, in a sense, the output of the system of Youssef \& Scoglio [1]. It is known that given sufficient training data, a neural network could approximate any function. In case of complex problems (such as the one considered in this paper), and for real experimental scenarios, the model might fail to achieve high levels of approximation accuracy. Thus, we explicitly provide the model with the physical characteristics underlying the problem in the form of the system of Youssef \& Scoglio [1]. Thus, by combining the neural network with the system of ODEs, the flexibility of the model increases, and can potentially more accurately predict the spread of epidemics. This is also confirmed by the improved performance of the proposed GN-ODE against the plain system of ODEs and the GNN variants.

---

> > ### Author Response · Authors · 2023-05-08
> > **Reply to Reviewer eMqf**
> >
> > **[R]: Time Comparisons:**
> >
> > Even though we agree with the reviewer that the comparison is not entirely fair, we should mention that the proposed model needs to be trained once, even on small networks, and can then be applied to any network. Importantly, the results of Figure 6 suggest that the proposed model exhibits good generalization ability and can produce accurate predictions even for large unseen networks. Arguably, from the perspective of a practitioner, what matters the most is the inference time and not the training time. Additionally, studies that include comparisons between algorithms and neural networks, report inference times of the relevant methods [3], since neural networks' training time depends on several parameters, such as the availability of GPUs, the hidden dimension size, the number of epochs, etc. A large pre-trained model could be applied to any network of interest. With regards to the inference time, in Figure 2(b), we observe that the proposed GN-ODE model is approximately 7 times faster than DMP on the largest of the considered datasets (Epinions). Also, in case the model is trained on small networks and is then applied to a larger network, its overall training time could potentially be smaller than the running time of DMP (running time depends on the number of epochs).
> >
> > **[R]: Others:**
> >
> > As requested by the reviewer, we have updated Figure 6 and we now also report the generalization performance of DMP to unseen networks. Furthermore, we have added a new Figure in the Appendix (Figure 8) which serves as a complementary Figure for the out of distribution experiments of parameters $\beta, \gamma$ of Figures 4 and 5.
> >
> >
> > [1] Sharkey, K. J. (2008). Deterministic epidemiological models at the individual level. Journal of Mathematical Biology, 57, 311-331.\
> > [2] Youssef, M., and Scoglio, C. (2011). An individual-based approach to SIR epidemics in contact networks. Journal of theoretical biology, 283(1), 136-144.\
> > [3] Kool, W., Van Hoof, H., and Welling, M. (2018). Attention, learn to solve routing problems!. arXiv preprint arXiv:1803.08475.

---

### Review · Reviewer_Ex2D · 2023-05-05

**Summary Of Contributions:**

The paper implements the well-known SIR model into a dynamic graph network pipeline, the dynamics of which are governed at a latent state space by a set of differential equations imported from the epidemiology domain. The paper demonstrates that in a number of network modeling data sets that the proposed method outperforms the state of the art either in prediction accuracy or in compute time.

**Audience:**

Yes

**Claims And Evidence:**

No

**Requested Changes:**

 - The relationships of beta/gamma parameters and a sample being in or out of distribution is not clear to me, also from the text. Are they not simply trainable parameters? What is the reason that makes them exogeneous?

- I am having hard time to interpret the OOD results reported in Figure 5. What is the take-home message there and how one should conclude from there that the proposed method is more advantageous than the baseline?

- Same for Figure 6. Why would one expect from the proposed model to work better in the small sample regime? What I see is quite the contrary. While GCN was not competitive in Figure 2, it all of a sudden comes to the level of GN-ODE here. What should we learn from this?

**Strengths And Weaknesses:**

Strengths:
  - The  idea is a rather straightforward combination of latent NODEs and graph neural nets. However, the application area is interesting and it has significant societal impact.
  - The building blocks constituting the model are well justified and they make a meaningful whole.
  - The reported main results are strong.

Weaknesses:
  - The significant dependency of the model performance on beta and gamma is a bit worrisome.
 - The connection of the rationale of the reported side experiments, e.g. OOD and small sample case to the main story line is not obvious.

---

> ### Author Response · Authors · 2023-05-08
> **Reply to Reviewer Ex2D**
>
> We would like to thank the reviewer for pointing out some details concerning the experimental evaluation and the proposed system that may need additional clarification.
>
> **[R]: Infection and Recovery Rates:**
>
> There has been a misunderstanding. Parameters $\beta$ and $\gamma$ are not trainable parameters of the neural network.
> They are instead parameters related to the spreading process and take fixed values for each instance of the SIR model on a network.
> Specifically, the parameters $\beta_{ij} \in [0,1]$ denote the infection rate of edge $(i \rightarrow j)$ and the parameters $\gamma_i \in [0,1]$ denote the recovery rate of node $i$. These parameters in terms of the SIR model constitute constants that determine the evolution of the stochastic process on the arbitrary networks of contacts.  The value of parameter $\beta$ determines the probability of a node moving from the susceptible set $S$ to the infected $I$, whereas parameter $\gamma$ represents the probability of recovering, i.e, moving to set $R$ after being infected. As described in Section 3, to create an instance for our problem, we randomly sample parameters $\beta$ and $\gamma$ and then generate the ground truth labels at each time step via Monte-Carlo simulations. This line of work is also followed in several other studies on the SIR model [1], where real-world data is not available and there is a need for obtaining the true values with simulations based on some given exogenous variables $\beta$, $\gamma$ and some initial conditions for the infected nodes $I_0$.
>
> **[R]: Out of distribution results:**
>
> In the initial within distribution experiments, after the generation of the samples via Monte-Carlo simulations, splitting into the train, validation and test sets was performed randomly, meaning that samples in the test set could potentially be described by any values of parameters $\beta$ and $\gamma$. Thus, those values could be close to the ones of some training sample(s).
> In this setting, we observe that the GCN model achieves slightly better performance than the proposed GN-ODE model on 2 out of the 8 studied networks. To investigate the generalization properties of the different methods, we present their performance in an additional out of distribution experimental setup.  We, therefore, select most samples to belong to the test set so that their parameters $\beta$ or $\gamma$ are different from the values of the corresponding parameters of instances contained in the training set.  For this set of experiments, we report the performance of the different methods in Figures 3a and 3b. The results demonstrate that the proposed method actually outperforms the baseline GNNs in terms of generalization performance. For the same experimental setup, we show in Figures 4 and 5, the loss achieved per sample in the test set of the different networks, including both samples with parameters within the ranges of parameters seen during training (i.e., inside the area of the vertical dotted grey lined), as well as unseen ones (i.e., outside those lines).
> Samples within the range (i.e., inside the area) receive lower errors for all GNN methods as expected, whereas for the unseen samples (i.e., outside the area) the proposed GN-ODE outperforms the baselines, showing low errors, which are comparable with the ones within.
>
> **[R]: Generalization to unseen networks:**
>
> In Figure 6, we provide the performances of the networks when trained on smaller networks and evaluated on larger unseen networks (that do not occur in the training set), contrary to the previous settings where we used the same network structure to train and evaluate the models. We also compare the former performances (i.e., training on multiple smaller networks) to the latter ones (i.e., training on the same single network) for the within distribution setting (shown in Figure 2 in which GCN outperforms the rest of the GNNs in Wiki-vote and Epinions).  The takeaway message from this experiment is that the proposed method keeps comparative performance to unseen networks even when not trained on instances of the network structure included in the test set, contrary to other GNN baselines that do not generalize well.
>
> [1] Shrestha, M., Scarpino, S. V., and Moore, C. (2015). Message-passing approach for recurrent-state epidemic models on networks. Physical Review E, 92(2), 022821.

---

### Review · Reviewer_b9Kg · 2023-05-17

**Summary Of Contributions:**

The submission presents a new model called the Graph Neural ODE (GN-ODE) model, which combines graph neural networks (GNNs) and ordinary differential equations (ODEs) to better capture dynamic relationships between nodes and improve prediction accuracy in epidemic spread. The paper also proposes a new message passing mechanism for forming the time-discretized approximation equation of the ODE solver. The paper evaluates the GN-ODE model on multiple datasets and compares it with other existing methods, demonstrating its high accuracy and robustness in predicting epidemic spread. The paper also provides in-depth analysis of how the GN-ODE model captures dynamic relationships between nodes and enhances its representational power using message passing mechanisms. Overall, the submission presents a novel and effective approach to predicting epidemic spread and contributes new knowledge to the field of modeling spreading processes.

**Audience:**

Yes

**Broader Impact Concerns:**

The paper does not appear to raise any significant ethical concerns that would require adding a Broader Impact Statement.

**Claims And Evidence:**

Yes

**Requested Changes:**

1.	A simpler system of differential equations was employed which consists of 3^n equations instead of 3n. The authors may need to mention the rationality of simplification and more detailed explanations of the used ODEs.
2.	The authors mentioned that prior knowledge in the form of a system of ODEs effectively improved the correctness of the function approximation. If there is a better ODEs model for epidemic dynamics, can it improve the algorithm performance more effectively? A small discussion is needed.
3.	The authors considered the message passing mechanism used to form the time-discretized approximation equation of the ODE solver and thought it is the reason why these models have demonstrated great success in several problems. Can you give more specific instructions of why this conclusion might be reached?
4.	The values of hyperparameters \beta and \gamma of SIR were randomly sampled from [0.1, 0.5], why was not the dataset generated from a larger range of these hyperparameters such as [0, 1]?
5.	The captions of Figure 4 and Figure 5 are exactly the same, which makes it impossible for people to understand the difference between them directly.

**Strengths And Weaknesses:**

Strong aspects:
1.	The proposed GN-ODE model is a novel and effective approach to predicting epidemic spread, which combines GNNs and ODEs to better capture dynamic relationships between nodes and improve prediction accuracy.
2.	The paper provides in-depth explanation of how the GN-ODE model captures dynamic relationships between nodes and enhances its representational power using message passing mechanisms.
3.	The paper evaluates the GN-ODE model on multiple datasets and compares it with other existing methods, demonstrating its high accuracy and robustness in predicting epidemic spread.
Weaker elements:
1.	The paper could benefit from more detailed explanations of some technical aspects, such as more analysis of the ODEs which is employed to model the system, and the message passing mechanism used to form the time-discretized approximation equation of the ODE solver.
2.	The captions of figures and the description of results should be clearer.

---

> ### Author Response · Authors · 2023-05-23
> **Reply to Reviewer b9Kg**
>
> We want to thank the reviewer for the valuable comments provided on our work. We answer to the requested changes in the following paragraphs.
>
> **[R]: Employed System of ODEs:**
>
> We have provided more details about the system of ODEs in the revised version of the manuscript (Section A.2 in the Appendix). We also provide performance comparisons between the system of ODEs and the proposed GN-ODE in the revised manuscript (Section A.3 in the Appendix). A detailed description of how the employed system of ODEs is derived is given in the response to reviewer eMdf (paragraph **''The fixed ODE system as an approximation''**). With regards to the reviewer's comment about the choice of the approximated ODE system, we should mention that the main attractive properties of the employed system of ODEs are that it is closed and it is more efficient than other competing models. Different systems of ODEs that better approximate the exact system would reach higher levels of performance compared to the employed system. Integrating such a system into the proposed architecture could further boost its predictive performance. However, more complex systems are computationally more expensive, and applying such a system to large networks might not be feasible. We thus believe that the employed system of ODEs achieves a good trade-off between performance and efficiency, and that the neural network module can learn to refine its output, making the whole approach both efficient and accurate.
>
> **[R]: Message Passing and Physical Systems:**
>
> We observe in Equation (1) that to update the representations of the nodes, the following matrix multiplication is performed: $\mathbf{A} \, \mathbf{I}_h$ where $\mathbf{A}$ is the adjacency matrix of the graph and $\mathbf{I}_h$ is a matrix that stores node features. This Equation which is derived from the employed system of ODEs essentially describes a message-passing scheme where nodes update their representations by aggregating the representations of their neighbors. This is also the main underlying idea behind most graph neural networks (GNNs). An interesting observation of our study is that physical systems on graphs are in several cases fundamentally described by equations involving message-passing operations [4,5]. As discussed in the manuscript, this connection between the message-passing mechanism of GNNs and the underlying dynamics of several complex processes could potentially explain the relatively good performance of such models in problems encountered in physics (such as the one considered in this study).
>
> **[R]: Choice of Infection and Recovery Rates Ranges:**
>
> The range for the infection and recovery rates for the SIR models was selected so as to imitate epidemic dynamics for real-world systems [1]. Very large values for the epidemic parameters (i.e. over 0.5) have experimentally shown to enforce an abrupt movement of several nodes to the infected or recovery sets, which practically in real-world epidemic scenarios does not happen [2,3]. This choice would not let us study the long-term evolution of epidemics in a wide range of network sizes as well as the effect of the graph structure on epidemics spreading. At the same time, very small parameter values (i.e. less than 0.1) significantly slow down the spreading process and do not allow us to obtain a clear picture of the models' performance on large networks (e.g., Enron and Epinions). Therefore, the hyperparameters were sampled from those intervals in order to create a realistic setting and to evaluate how useful each method could be in a real-world scenario.

---

> ### Author Response · Authors · 2023-05-23
> **Reply to Reviewer b9Kg**
>
> **[R]: Figures:**
>
> We would like to thank the reviewer for pointing this out. These two Figures could be actually merged into a single Figure, but we have chosen to provide 2 separate Figures due to formatting issues. The 8 subfigures (4 subfigures of Figure 1 and 4 subfigures of Figure 2) illustrate the performance of the model and the baselines on the 8 considered datasets (one subfigure for each dataset). Since we study the impact of the different values of both $\beta$ and $\gamma$ on the models' performance, we would need 16 subfigures in total (2 subfigures for each dataset). However, due to space constraints, we provide only one of them for each dataset (performance as a function of either $\beta$ or $\gamma$).
>
> [1] Marinov, T. T., and Marinova, R. S. (2022). Adaptive SIR model with vaccination: Simultaneous identification of rates and functions illustrated with COVID-19. Scientific Reports, 12(1), 1-13.
> [2] Kitsak, M., Gallos, L. K., Havlin, S., Liljeros, F., Muchnik, L., Stanley, H. E., and Makse, H. A. (2010). Identification of influential spreaders in complex networks. Nature physics, 6(11), 888-893.
> [3] Melikechi, O., Young, A. L., Tang, T., Bowman, T., Dunson, D., and Johndrow, J. (2022). Limits of epidemic prediction using SIR models. Journal of Mathematical Biology, 85(4), 36.
> [4] Karrer, B., and Newman, M. E. (2010). Message passing approach for general epidemic models. Physical Review E, 82(1), 016101.
> [5] Weigt, M., White, R. A., Szurmant, H., Hoch, J. A., and Hwa, T. (2009). Identification of direct residue contacts in protein–protein interaction by message passing. Proceedings of the National Academy of Sciences, 106(1), 67-72.

---

### Author Response · Authors · 2023-05-08
**Summary of Revisions**

We thank the reviewers for their constructive reviews. We have updated the paper to meet the requested changes.

Here is the summary of modifications made in the revisions:
- **[Subsection 3.2.2 - Spreading Prediction on Multiple Networks]** DMP performance included in Figure 6.
- **[Section A.2 of the Appendix]** More detailed description of the employed system of ODEs for the SIR model and its relation to the exact system.
- **[Section A.3 of the Appendix]** Comparison of the proposed GN-ODE model against the Fixed System of ODEs (Table 2).
- **[Section A.4 of the Appendix]** Further Figures for the out of distribution generalization experiments.
- **[Subsection 3.2.1 - Spreading Prediction on a Single Networks]** Updated captions of Figures 4 and 5.
- **[Subsection 3.2.1 - Spreading Prediction on a Single Networks (Page 6)]** Provided more details about how the values of hyperparameters $\beta$ and $\gamma$ were chosen.

---

> ### Comment · Reviewer_eMqf · 2023-06-21
> **Thanks for the improvements in the revision!**
>
> This revision is significantly improved in my opinion. I think the paper was already quite strong with high novelty and potential impact. The main improvement in the revision is in the presentation and explanation about the system of ODEs.

---

### Decision · Action_Editors · 2023-07-21

**Recommendation:** Accept as is

**Comment:**

All the reviewers agree that the contributions are novel (although combination of graph neural networks and neural ODEs is quite straightforward, there is no too much literature on that).
The combined model for epidemics is surely novel and has not been considered before.
The numerical evidence confirms that the method is effective.

The paper can also inspire the usage of neural odes in epidemics community,, which can be potentially very impactful, thus this submission can be certified as "featured".

**Audience:**

I think the potential audience for this paper is quite broad: it includes both experts in graph neural networks and ODEs and experts in more applied topics such as epidemics (and more general, network science) modelling.

**Claims And Evidence:**

The paper presents graph neural odes and applies them for modelling epidemic spreading.

1. The combination of graph neural networks and neural odes is a novel idea.
2. The application is interesting and practical